# FLASH Radiotherapy: Expectations, Challenges, and Current Knowledge

**DOI:** 10.3390/ijms25052546

**Published:** 2024-02-22

**Authors:** Andrea Borghini, Luca Labate, Simona Piccinini, Costanza Maria Vittoria Panaino, Maria Grazia Andreassi, Leonida Antonio Gizzi

**Affiliations:** 1CNR Institute of Clinical Physiology, 56124 Pisa, Italy; mariagrazia.andreassi@cnr.it; 2Intense Laser Irradiation Laboratory (ILIL), CNR Istituto Nazionale di Ottica, 56124 Pisa, Italy; luca.labate@ino.cnr.it (L.L.); simona.piccinini@ino.cnr.it (S.P.); costanzamariavittoria.panaino@ino.cnr.it (C.M.V.P.); leonidaantonio.gizzi@ino.cnr.it (L.A.G.)

**Keywords:** FLASH radiotherapy, FLASH effect, ultra-high dose rate, very high-energy electrons, normal tissue response, tumor response, nuclear DNA damage, γ-H2AX, CBMN assay, mitochondrial DNA

## Abstract

Major strides have been made in the development of FLASH radiotherapy (FLASH RT) in the last ten years, but there are still many obstacles to overcome for transfer to the clinic to become a reality. Although preclinical and first-in-human clinical evidence suggests that ultra-high dose rates (UHDRs) induce a sparing effect in normal tissue without modifying the therapeutic effect on the tumor, successful clinical translation of FLASH-RT depends on a better understanding of the biological mechanisms underpinning the sparing effect. Suitable in vitro studies are required to fully understand the radiobiological mechanisms associated with UHDRs. From a technical point of view, it is also crucial to develop optimal technologies in terms of beam irradiation parameters for producing FLASH conditions. This review provides an overview of the research progress of FLASH RT and discusses the potential challenges to be faced before its clinical application. We critically summarize the preclinical evidence and in vitro studies on DNA damage following UHDR irradiation. We also highlight the ongoing developments of technologies for delivering FLASH-compliant beams, with a focus on laser-driven plasma accelerators suitable for performing basic radiobiological research on the UHDR effects.

## 1. Introduction

The prevention or mitigation of radiation-induced damage to normal tissues has always been a theme of interest in radiotherapy research. Ongoing studies are focusing on developing new treatment modalities aiming to reduce the risk of complications arising from radiation treatments. FLASH radiotherapy (FLASH RT) is one of the most promising approaches based on the normal tissue-sparing effects of ultra-high dose rate (UHDR) irradiations [1].

FLASH RT is based on the delivery of UHDR radiation several orders of magnitude higher than what is presently used in clinical conventional radiotherapy (CONV RT) (≥40 Gy/s vs. ≤0.03 Gy/s) [2,3]. Even though FLASH RT has been defined using its mean dose rate, the complete definition requires other physical parameters, such as the repetition rate, number of pulses, and the total duration of irradiation. Moreover, the FLASH effect is most thoroughly characterized by electron irradiations, but proton and X-ray UHDR irradiations have been shown to reduce toxicity in healthy tissues, maintaining a similar tumor control compared to CONV RT [4,5,6].

FLASH RT has potential benefits corroborated by a growing body of preclinical data [7,8,9]. Once the potential of FLASH RT will be confirmed in clinical trials, this novel technology may revolutionize the field of radiation oncology, becoming the principal modality of radiotherapy for certain tumors [10]. In view of this exciting perspective, more research is needed to better understand the conditions inducing the FLASH effect. 

Several mechanisms have been proposed and new hypotheses are emerging to explain the dose rate-dependent differential response in healthy tissues [4,5,6]. However, despite several steps forward, these theories remain yet to be fully confirmed. This is a mandatory step for a successful clinical translation of FLASH RT. 

In this review, we summarize the research progress of FLASH RT and the current theories explaining the FLASH effect, highlighting issues and future considerations.

## 2. FLASH Radiotherapy: Tumor and Normal Tissue Responses

Hints of the FLASH effect were first observed in 1959. When bacteria were exposed to UHDRs (10–20 krads/2 μs), radiosensitivity was reduced compared to the conventional dose rate irradiation [11]. Similar results were also found in mammalian cells in later studies [12,13]. However, the research on the FLASH effect in the 1960s and 1970s was not translated into clinical applications and stagnated until its recent resurgence. Over the last few years, in fact, a growing body of studies pointed to the potential capacity of FLASH RT in different tissues using different preclinical models.

### 2.1. Lung Tissue

In 2014, a well-established mouse model of lung fibrosis was presented as the first proof-of-principle study [1]. A significant reduction in normal tissue injury was identified with electron FLASH RT [1,14], while the overall treatment efficacy did not appear to differ at similar doses compared to CONV RT. FLASH irradiation showed a protective effect against pneumonia and fibrosis at a dose of 17 Gy compared to conventional dose rate irradiation. However, at a higher dose of 30 Gy, mice subjected to FLASH irradiation began to develop pneumonia and fibrosis [15]. 

The potential benefits of UHDRs from proton beams have been also investigated in a mouse model of non-small-cell lung cancers, receiving thoracic radiation therapy using CONV RT (<0.05 Gy/s) and FLASH RT (>60 Gy/s) [16]. FLASH dose rate proton delivery was shown to modulate the immune system, improving tumor control. In particular, proton FLASH RT was more efficient compared to CONV RT in increasing the infiltration of T-lymphocytes inside the tumor, simultaneously reducing the percentage of immunosuppressive regulatory T-cells. Moreover, FLASH RT was more effective in reducing pro-tumorigenic M2-like macrophages and the expression of checkpoint inhibitors in the tumor, indicating a decreased immune tolerance [16].

Recently, the FLASH effect has been produced in both single and fractionated irradiation (ten pulses), with a dose of 2 Gy as the minimum dose to obtain the FLASH effect [17]. Regarding the effects of both FLASH schemes on normal lung tissue, the pulmonary pathology was similar, and only some inflammatory cells were observed. Slight thickening of the alveolar septum and interstitial hemorrhage were identified instead after CONV RT [17].

### 2.2. Brain Tissue

The most extensive data about FLASH RT for the central nervous system arises from Montay-Gruel and colleagues. In 2017, they first performed preclinical studies on the brain tissues of mice, demonstrating that spatial memory was significantly protected with an average dose rate of radiation >100 Gy/s. Even 2 months after irradiation, the ability of mice to recognize objects was significantly better after electron FLASH RT compared to CONV RT. Interestingly, the protective effect of FLASH RT on nerve regeneration depended on the protective effect of neural stem cells [18]. 

Further research in experimental models with intracranial tumors was performed in the succeeding years. Altogether, these studies concluded that FLASH RT had a more considerable protective effect on healthy brain tissue than conventional dose rates [19,20,21,22,23].

To approximate clinical treatment scenarios, hypofractionated electron FLASH RT has been proposed as an effective treatment against glioblastoma. Mice that received FLASH RT, either as a 10 Gy single dose or hypo-fractionated regimens (2 × 7 Gy and 3 × 10 Gy), exhibit neurocognitive sparing, maintaining the same efficiency as CONV-RT in delaying tumor growth [20]. 

These results were later confirmed by Allen et al. [24]. In their elegant work, the authors used a radiosensitive juvenile mouse model exposed to hypofractionated (2 × 10 Gy, FLASH-RT or CONV-RT) radiotherapy in order to assess adverse long-term neurological outcomes. Hypofractionated electron FLASH RT entailed significant and long-term healthy tissue protection in the mouse brain. FLASH RT preserved synaptic plasticity and integrity, reduced neuroinflammation, and preserved the cerebrovascular structure [24].

In 2018, Montay-Gruel et al. proposed the FLASH effect triggered by X-rays [18]. A 10 Gy whole-brain irradiation delivered at high dose rates (37 Gy/s) with synchrotron-generated X-rays prevented brain damage, with a better preservation of hippocampal cell division and a decrease in reactive astrogliosis compared to X-ray irradiation performed at a conventional dose rate [18]. These results were fully comparable with their previous results obtained with electron FLASH RT. 

Studies on the FLASH effect in the brain tissue with electron and photon irradiations show FLASH benefits in the preclinical mouse model, but data are limited for protons. It is only recently that experimental evidence of neuroprotection has been gathered using proton beam irradiation [25]. In mice, cranially irradiated with proton FLASH or CONV dose rates at a single dose of 25 Gy, FLASH RT was found to spare memory impairment and induce a similar tumor-infiltrating lymphocyte recruitment [25].

### 2.3. Skin Tissue

Numerous preclinical studies investigated the FLASH effect in terms of reduced skin toxicity in mice using UHDR proton and electron irradiations [26,27,28,29]. Notably, Zhang et al. investigated the protective role of FLASH proton irradiation (130 Gy/s) on the skin varying the oxygen concentration. FLASH proton irradiation decreased skin contraction, epidermis thickness, and collagen deposition compared to conventional irradiations. Interestingly, this effect was controlled by changing oxygen concentration, highlighting the role of oxygen in the FLASH tissue-sparing effect. In fact, FLASH tissue sparing was not observed for mice breathing pure oxygen for 6 min pre-irradiation until after the irradiation was completed. Hypoxic skin also did not result in a difference in outcome between FLASH and conventional dose rate irradiations [29]. To prompt the clinical transfer, Vozenin and colleagues assessed the FLASH effect in higher mammals, including in minipigs and cats [30]. Using the radiation-induced depilation and skin fibrosis as acute and late endpoints, respectively, a protective effect of FLASH-RT was observed in minipigs and cats [30]. Pig skin was irradiated to single-fraction radiation doses of 28, 31, or 34 Gy using either CONV (0.083 Gy/s) or UHDR electron irradiation (300 Gy/s). The presence of late effects, such as fibronecrosis, collagen deposition, and skin contracture, was greater in animals irradiated with CONV dose rates [30]. These preclinical results are consistent with results obtained in a veterinarian clinical trial conducted in cat patients with squamous cell carcinoma of the nose. Cats were treated using single fractions from 25 to 41 Gy at ultra-high dose rates. No dose-limiting toxicity and relatively mild long-term toxicity were found. FLASH-RT treatment yielded a favorable outcome with complete response at 3 months for all cat patients and a free survival rate of 84% at 16 months [30].

While these preliminary studies demonstrated the potential advantage of FLASH RT, the results obtained later by the same group underscored the potential limitations of FLASH RT, emphasizing the need for caution and additional investigations [31]. Indeed, a randomized phase III trial was conducted to investigate FLASH RT in cats with spontaneous tumors with a long-term follow-up. In parallel, the sparing ability was also studied on minipigs. Surprisingly, the trial was prematurely interrupted due to maxillary bone necrosis, which occurred 9 to 15 months after radiotherapy in three of seven cats treated with FLASH-RT (43%) compared with zero of nine cats treated with CONV-RT. In pigs, even though no acute toxicity was recorded, severe late skin necrosis occurred in a volume-dependent manner [31]. 

Promising data at the skin level emerge also from the canine model, where superficial solid cancers (melanomas, squamous cell carcinomas, soft tissue sarcomas, and mast cell tumors) were irradiated with 15–35 Gy FLASH-RT. Adverse events observed after a follow-up of 3–6 months were mild and consisted of local alopecia, dry desquamation, leukotricia, and mild erythema. Regarding the tumor response, eleven out of thirteen irradiated tumors showed partial or complete response [32].

### 2.4. Intestine Tissue

Several studies have investigated the FLASH effect on the intestine [33,34,35,36,37,38]. Electron FLASH has been shown to reduce changes in microbiota with UHDR of about 280 Gy/s at doses of 7–12 Gy [34]. Moreover, both proton and X-ray FLASH irradiation spare mouse intestinal crypts [35,37]. Interestingly, Diffenderfer et al. designed and dosimetrically validated a proton FLASH RT system with accurate control of beam flux on a millisecond timescale and online monitoring of the integral and dose delivery time structure. Utilizing this system, the authors first demonstrated that whole abdominal proton FLASH RT (78 ± 9 Gy/s) reduced acute cell loss and late fibrosis following both whole-abdomen and focal intestinal treatments while maintaining comparable tumor growth inhibition between the two modalities [38]. Proton beams were also found not to induce the sparing effect [33]. Partial abdominal FLASH irradiation (~120 Gy/s) delivered to C57BL/6j and immunodeficient Rag1-/-/C57 mice has been shown not to spare intestinal tissue or circulating blood lymphocytes [33]. There was no difference in the number of lymphocytes between FLASH and CONV RT; a similar number of proliferating crypt cells and thickness of the muscularis externa were found [33]. However, as stated by the authors, comprehending the variances between their settings (partial gut, 120 Gy/s proton irradiations, 14–17 Gy) and those employed at other institutes [36,38], which demonstrated FLASH tissue sparing in the gut, may aid in determining the optimal FLASH tissue-sparing conditions. In fact, it is conceivable, and perhaps probable, that total gut or total abdominal irradiation may induce a FLASH effect, whereas partial-gut irradiation may not.

### 2.5. Blood Tissue

Most of the experimental studies were performed using models of whole organ irradiation; inversely, the impact on blood tissue at the UHDR has been only recently investigated [39]. Using a prototype 6 MeV electron beam linear accelerator, the effect of FLASH total body irradiation was analyzed on humanized models of T-cell acute lymphoblastic leukemia. In particular, three T-ALL patient-derived xenografts and hematopoietic stem and CD34+ cells isolated from umbilical cord blood were transplanted into immunocompromised mice. Mice were irradiated with 4 Gy FLASH and CONV, and tumor growth and normal hematopoiesis were assessed. Interestingly, FLASH RT reduced functional damage to human blood stem cells and presented a therapeutic effect on human leukemia [39]. Jin and colleagues [40] employed a dose rate-dependent model to assess the quantity of circulating immune cells in the bloodstream. Their study examined the influence of the radiation dose rate on the impairment of circulating immune cells, revealing a decrease in immune cell depletion as the dose rate increased. Notably, this effect was subject to dose/fraction retention, with a more significant impact observed at doses/fractions ranging from 30 to 50 Gy, diminishing at 5 Gy, and being almost negligible at 2 Gy. A decrease in the mortality of circulating immune cells from 90 to 100% at conventional dose rates to 5–10% at UHDRs was observed [40].

Recently, the effects of FLASH RT on blood lymphocytes in humans and small animals were analyzed using a mathematical model [41]. This model has been developed to depict the survival level of lymphocytes in the bloodstream following FLASH RT and lower dose rates of partial-body irradiation. This model is expressed through analytic formulae, incorporating several parameters, such as physiological factors (blood flow characteristics), biophysical factors (lymphocyte radiosensitivity), and physical parameters related to irradiation. It has been observed that FLASH irradiation in humans, administered at doses ranging from 10 to 40 Gy and exposure durations significantly shorter (<1 s) than the blood circulation time (∼60 s), results in maximal blood lymphocyte sparing. For the specified dose range, the modeling predicts that effective dose rates for optimal blood lymphocyte sparing in humans fall within the range of ≥40 Gy/s, coinciding with the dose rate range employed in FLASH radiation therapy. Moreover, it has been identified that the effective dose rates for mice are higher than those for humans (within the same dose range) due to the shorter blood circulation time in mice compared to humans [41].

In a more recent compelling study conducted by Galts et al., the authors set out to construct a dosimetric framework to assess the potential advantages of conformal proton beam scanning FLASH therapy. Their focus was on evaluating the sparing of circulating blood lymphocytes and comparing it with hypofractionated FLASH schemes, along with conventional fractionated intensity-modulated proton therapy treatment plans [42]. Interestingly, the FLASH effect manifested in circulating lymphocytes. Specifically, FLASH radiotherapy (RT) resulted in a 69.2% reduction in the depletion rate of lymphocytes compared to conventional fractionated RT [42].

### 2.6. Zebrafish as an Emerging Model System

Zebrafish is emerging as an intriguing model to investigate the FLASH effect. The feasibility of electron FLASH RT has been tested with positive results in zebrafish embryos. In fact, electron beams showed fewer morphological alterations than CONV RT at doses above 10 Gy [3]. No significant impact of high proton dose rates was shown for embryonic survival, and the rate of spinal curvature was one type of developmental abnormality. For the rate of pericardial edema as an acute radiation effect, a significant reduction after proton FLASH RT (100 Gy/s) was also observed [43].

Karsch et al. included FLASH dose rates using electrons and protons resembling isochronous cyclotrons, synchrocyclotrons, and synchrotrons. The sparing effect was observed in the zebrafish embryos and resulted in being dependent on the mean dose rate and radiation time [44]. Low partial oxygen levels also appeared to have a stronger FLASH effect compared to high partial oxygen levels [45].

In a zebrafish embryo model, the first evidence of the in vivo FLASH effect with helium ions has been recently reported [46]. UHDR helium ions spared body development and reduced spine curvature compared to the conventional dose rate, advocating for the validity of combining high LET ion beams with UHDR modality to take advantage of both good ballistics and reduced toxicity [46].

## 3. Treatment of Human Patients and First Clinical Trials

The first human patient with refractory cutaneous lymphoma was treated with electron FLASH RT in 2018 [47]. FLASH RT treatment was given with a 5.6-MeV linac (Oriatron, PMB Alcen, France) and resulted in being practicable with a positive outcome on the tumor and normal skin [47]. Regarding the skin surrounding the tumor, there was no decrease in the epidermal thickness and disruption at the basal membrane. An asymptomatic mild epithelitis and a grade 1 edema were found at 3 weeks. However, for this patient, when compared to previous skin reactions after exposure to 20 Gy in ten fractions or 21 Gy in six fractions, the FLASH RT adverse effects were minimal and disappeared in a much shorter time. Similarly, the tumor response was durable, with a follow-up of 5 months. Of interest, this patient was subsequently treated for two additional tumors with FLASH and conventional dose rates (166 Gy/s vs. 0.08 Gy/s). At the dose level of 15 Gy, ultra-high and conventional dose rates had similar tumor control along with similar acute and late toxic effects [48]. Although the effects of FLASH and CONV appeared similar, the limitations of this study (its case report nature, the utilization of only a single dose level, and the inability to conduct statistical testing of a null hypothesis) prevent the authors from conclusively ruling out the possibility that the effects of the two types of treatment may still differ [48].

The protocol for the first-in-human clinical investigation of proton FLASH RT has been recently described by Daugherty et al. [49]. FLASH radiotherapy for the treatment of symptomatic bone metastases (FAST-01) is a prospective, single-center trial (NTC04592887) designed to evaluate the efficacy and toxicity of palliative FLASH treatment of bone metastases [49]. The study demonstrates that proton FLASH treatment (dose rate = 51–61 Gy/s, single dose = 8 Gy) was clinically practicable in the treatment of bone metastases, with the efficacy and the presence of adverse effects being analogous to CONV RT [50]. The main results of the trial indicated that eight out of the twelve treated sites experienced complete or partial pain following FLASH treatment.

In the patient group undergoing treatment, twelve adverse events were reported. The majority of these events, specifically eleven out of twelve, were classified as grade 1 adverse events, such as skin hyperpigmentation, edema, erythema, fatigue, and pruritus. Additionally, there was one patient who experienced a grade 2 adverse event, specifically extremity pain, one month after the treatment [50]. Additional phase I (NCT04986696, NCT05524064) and phase II (NCT05724875) clinical trials, aimed at describing and comparing the toxicity and efficacy of high dose rates, are ongoing and will recruit patients with skin melanoma metastases, bone metastasis in the thorax and cutaneous squamous, or basal cell carcinoma [51].

Figure 1 summarizes the preclinical and first-in-human clinical evidence about the FLASH effect. These studies show that, to date, the capability of FLASH RT to spare healthy tissues has been investigated in several tissues using preclinical models of different genetic backgrounds. Further data at different experimental conditions (e.g., dose, dose rate, oxygen tension) are necessary to confirm the protection of normal tissue under FLASH RT. 

## 4. Biological Mechanisms behind the FLASH Effect: The Role of DNA Damage

Non-mutually exclusive hypotheses regarding the mechanism underlying the FLASH effect have been proposed, such as the rapid oxygen depletion and reactive oxygen species (ROS) production, DNA damage, and the immune and inflammatory processes [6,7,8]. However, even though one of the most widely considered hypotheses is that the effect is related to substantial oxygen depletion upon FLASH, recent observations showed that oxygen depletion during pulse irradiation at an ultra-high dose rate is marginal and cannot entirely account for the FLASH effect in healthy normoxic tissues [52,53,54].

It is well recognized that nuclear DNA is the primary target and the most crucial molecule in the response to radiotherapy [55,56]. The subsequent cascade of DNA damage response (DDR) and signaling pathways are essential in determining the fate of cancer cells, such as death or survival [55,56]. Thus, elucidating DNA damage associated with UHDR irradiation is the most crucial radiobiological mechanism in order to fully define the benefits associated with FLASH RT. 

The extent of ionizing radiation-induced DNA damage alterations primarily depends on the density dose, dose rate, and linear energy transfer (LET), which is a measure of locally absorbed energy (kiloelectron volts, keV) per unit length (micrometer, µm). Low LET photon (X-ray or γ rays) irradiation implies a homogenous deposition of energy throughout the tissue volume, whereas high LET radiation (protons, alpha particles, and heavy ions), decelerates faster than photons, leading to the formation of a rapid Bragg peak [55,56], with penetration depth in tissues increasing with the beam energy (Figure 2).

Both low and high LET radiation act directly or indirectly on the DNA target. The direct effects are induced by ionizations and excitations of DNA molecules directly, disrupting the molecular structure. The indirect effects are mediated by water radiolysis, and free radicals are produced, which act as intermediaries causing DNA damage.

Typically, radiolytic events occur in three main stages taking place on different typical timescales. During the first or “physical” stage, which takes place within 10^−15^−10^−12^ s, extremely reactive free radicals (e.g., aqueous or hydrated electrons and other reactive oxygen species, such as H_2_O_2_, O_2_^−^, or OH^−^) are produced and undergo fast reorganization in the chemical stage (10^−12^–10^−6^ s), leading to the formation of an array of reactive products, which, in turn, can break the chemical bonds and produce DNA damage and possible repair processes in the cell over a wide timescale (“biological” stage). FLASH irradiation is around 1000 times faster than conventional irradiation, and this might interfere with the radiation–chemical reactions, and, consequently, with the biological processes in response to irradiation. 

Radiation treatment can produce a wide variety of DNA lesions, such as double-strand breaks (DSBs), which are considered the most deleterious lesions, damaging and killing cancer cells and leading to an effective therapeutic effect of radiotherapy.

High LET radiation is more lethal than similar doses of low LET radiation types, which is probably a result of the condensed energy deposition pattern and a very dense ionization pattern, which induces highly condensed DNA damage and is considered highly complex damage that is more difficult to repair. Ultra-high dose rate irradiation may have a significant impact on the DNA damage and the DNA response compared to conventional ion beam effects due to both spatial and temporal differences in their delivery (Figure 2). Energy delivery is typically associated with only early-time physical interactions, such as ionizations and excitations, and it does not interact with the biochemical and biological steps [57].

Knowing how cells respond to DNA damage is critical for understanding the FLASH effect, and suitable in vitro studies are required to fully evaluate this damage. Several in vitro assays can be employed to quantify ionizing radiation-induced DNA damage from different radiation beams. Two of the most commonly used tests are the comet assay and the analysis of the phosphorylated histone variant (γH2AX) (Figure 3).

The comet assay can detect DSBs using neutral single-cell gel electrophoresis, whereas the alkaline single-cell gel electrophoresis is more sensitive for the detection of SSBs [58]. Even though the comet assay is a fast and easy method to evaluate the degree of DNA damage, it has limitations regarding specificity and sensitivity, such as a limited dynamic range [58]. γ-H2AX is a protein marker that is quickly phosphorylated at sites of DSB and, therefore, can be microscopically visualized as nuclear foci by immunofluorescence [59]. The analysis of γH2AX foci allows DSB detection even in the very low dose range, going down to a single cell [59]. The main disadvantage of this analysis is the highly dynamic change in γH2AX foci early after irradiation. Additionally, the loss of γ-H2AX foci is a reasonable indicator of the timescale of rejoining DSB induced by low LET radiation but is less appropriate for those induced by high LET radiation [60]. Cytogenetic tests are the golden standard in radiobiology to quantify radiation-induced DNA damage [61]. These tests include the chromosomal aberration analysis, especially dicentric chromosome formation and the cytokinesis block micronucleus assay (CBMN), which are considered the most sensitive and reliable DNA biomarkers (Figure 3). The gold standard technique is the dicentric chromosome assay due to its high specificity for radiation [61], but CBMN often remains the preferred approach as it has the important advantage of allowing an economical, easy, and quick analysis of chromosomal damage (chromosome fragments or whole chromosomes) [61].

To date, the number of studies on DNA damage following UHDR irradiation is limited. Using γH2AX, proton FLASH RT was shown to generate less DNA damage compared to CONV RT in both normal lung fibroblasts and lung progenitor cells [62,63]. FLASH-irradiated lungs presented a reduction in DNA damage and senescent cells, suggesting a higher potential for lung regeneration with FLASH [63]. 

These results are in contrast with other studies, which indicate a non-significant difference in the induction of γH2AX-foci between FLASH and CONV RT [64,65]. Similarly, no divergence in DSB induction was found after proton UHDRs (48.6 Gy/s) vs. conventional dose rates of 0.057 Gy/s [66]. However, FLASH RT was able to reduce non-clustered DSB damages, such as single-strand breaks (SSBs) [66]. 

In this regard, Ohsawa et al. [67] analyzed the rate of SSBs and DSBs in the plasmid DNA of aqueous conditions by proton beams under CONV and FLASH regimes. The SSBs were significantly reduced after FLASH RT; instead, DSB induction was only slightly less than CONV RT [68]. Thus, these findings demonstrate that the FLASH effect would be effective in reducing non-lethal damage. In fact, compared to DSBs, SSBs represent non-lethal damage that is easily repaired in the living cells. Using a DNA-based phantom containing plasmid DNA, FLASH irradiation has been shown to decrease both DSBs and SSBs [68].

Cooper et al. recently identified an ex vivo FLASH-sparing effect using the alkaline version of the comet assay [69]. This fascinating work has been the first to directly show that FLASH-induced DNA damage is modulated by oxygen tension, total dose, and dose rate, with FLASH inducing lower levels of DNA damage for doses > 20 Gy, dose rates > 30 Gy/s, and 0.5% O_2_. These findings clearly show that the presence of both induced hypoxia and lower DNA damage can contribute to normal tissue-sparing effects [69]. More recently, the same group lent further support to the transient oxygen depletion mechanism as a driver of the reduced damage burden mediated by FLASH [70]. Using a high-throughput genome-wide translocation sequencing approach, a very recent study found no significant differences in the decrease in translocations or alteration of junction structures between FLASH-RT and CONV-RT in that human embryonic transformed kidney cell line (HEK239T cells) across a wide range of oxygen tensions. This suggests that both modalities of dose delivery induce similar DNA damage responses in the in vitro model under investigation, independent of the concentration of oxygen [71].

Two studies addressed the impact of UHDRs on chromosomal DNA damage. Using the CBMN assay, the effects of both single and multiple electron pulses have been investigated over different dose rates per pulse (instantaneous dose rate) [72]. Lymphocytes were exposed to graded doses from 2 to 8 Gy at different dose rates per pulse, ranging from 1 × 10^6^ Gy/s to 3.2 × 10^8^ Gy/s. Interestingly, a significant decrease in the MN yield with increasing dose rates per pulse was observed when the dose was delivered by a single electron pulse. The decrease in MN yield at higher dose rates suggests possible radical recombination, which, in turn, reduces biological damage [72]. More recently our group investigated the radiobiological effectiveness of ultra-short laser-driven electron bunches through the analysis of chromosomal damage in human peripheral blood lymphocytes [73]. Laser-driven electron accelerators are capable of producing very high dose rate high-energy electron bunches in shorter distances than conventional radiofrequency accelerators. Using the CBMN assay, our data showed that electron pulses (~1.5 MeV electrons, instantaneous dose rate 10^11^–10^12^ Gy/s) were more effective in inducing micronuclei compared to 50 kV X-rays at a lower dose rate (~94 mGy/min). Indeed, the results indicated that the MN yield induced by electron irradiation was higher when an individual dose was compared with that of X-rays [73].

Generally, the effect of ionizing radiation has been considered to be mainly due to nuclear DNA damage and molecular mechanisms of the induction and signaling of DNA damage. However, other cellular components can also be affected by radiation-targeted and non-targeted effects, such as mitochondrial DNA (mtDNA) [74].

The potential role of mitochondria in mediating the FLASH effect has been recently highlighted [75,76]. The effect of proton FLASH RT (100 Gy/s) on mitochondrial function has been assessed in lung fibroblasts under an ambient oxygen concentration (21%). FLASH RT generated marginal mitochondrial damage in terms of morphological changes, functional changes (membrane potential, mtDNA copy number, and cellular ATP levels), and ROS production [75]. Another in vitro study revealed a decrease in apoptosis and necrosis in FLASH (>10^9^ Gy/s)-irradiated cyt c-/c- cells compared to cyt c+/+ cells in both normoxic and hypoxia conditions. The reduction in cyt c release, as a consequence of less mitochondrial damage, could partially be responsible for the FLASH effect [76]. 

Other plausible mechanisms could account for the FLASH effect, such as stem cell niche preservation, differential lipid peroxidation and Fenton chemistry, and changes in specific protein classes (cytoskeleton), which are known to differ between normal and tumor cells [2,4,5,6]. Epigenetic mechanisms and changes in chromatin structure may also play a potential role in mediating the FLASH effect and will require further experimental investigation. Therefore, additional targets and metabolic processes should be considered with current technologies to provide deeper mechanistic insights into the beneficial FLASH effect in normal tissue compared to the tumor.

In fact, additional information can be obtained by properly varying beam properties that, in principle, can act on the physical processes behind the FLASH effect. Here, we expect that parameters like radiation type, temporal and spatial structure, dose range, and energy spectrum are expected to play a role. Systematic studies of these dependencies are still lacking, but a number of studies are emerging that focus on specific radiation parameters. Among these parameters, the beam temporal structure was varied to investigate oxygen depletion in water [77], measuring O_2_ content for different average and bunch dose rates of electron beams, showing a strong correlation with biological data, and supporting the role of radicals at the origin of the FLASH effect. In another study [78], the dependence of O_2_ consumption and H_2_O_2_ production were found to depend on the mean dose rate, with instantaneous dose rates also contributing to this effect. Concerning the type of radiation, interesting studies are finally emerging on the use of UHDR kilovoltage (kV) X-rays from a rotating-anode X-ray source for in vivo studies, making this type of radiation easily accessible at a laboratory scale compared to synchrotron radiation [79]. Again, concerning the type of radiation, a comprehensive comparison of electron and proton irradiation with UHDR and CONV modalities showed that the neurocognitive capacity of both electron and proton FLASH-irradiated mice was indistinguishable from the control, while both electron and proton CONV-irradiated cohorts showed cognitive decrements [80]. More specifically, normal brain protection was achieved when a single dose of 10 Gy was delivered in 90 ms or less, suggesting that the most important physical parameter driving the FLASH-sparing effect might be the mean dose rate.

## 5. Technologies for FLASH Radiation Beams

The characterization of beam parameters and dosimetry to produce the FLASH effect is fundamental for the clinical translation of the UHDR RT. FLASH RT relies on a combination of dose, dose rate, and irradiation time that falls outside the operational range of existing conventional clinical linear accelerators. These accelerators deliver doses through beams of X-ray photons with a broad energy spectrum produced by the bremsstrahlung of primary electrons, typically with energies ranging from 6 to 20 MeV. Unfortunately, the conversion of electrons into photons is highly inefficient, significantly limiting the maximum dose rate achievable at the treatment crosshair. Achieving the required dose rate for FLASH RT would necessitate a power increase of a thousand times or more in the existing clinical linear accelerators. These circumstances are motivating major efforts in the scientific and technological development of accelerators, including upgrades to existing experimental devices or the design and construction of entirely new systems based on advanced and disruptive concepts. A review of all the ongoing developments in this field is beyond the scope of this paper, and we redirect the readers to comprehensive reviews [81,82]. According to these studies, we highlight that while a number of technological solutions exist for experimental UHDR radiation sources that are enabling radiobiological studies, a robust, compact, and viable solution for a clinical accelerator able to treat deep-seated tumors is not yet available and may require years of development and significant investments. For radiobiological studies at a small laboratory scale, high dose rates are nowadays easily provided by enhanced IORT-like machines, delivering UHDR beams in the 5–10 MeV range electron beam [83]. While currently being used for in vitro studies and proof-of-concept in vivo studies on small animals, in perspective, these devices will also enable the treatment of skin tumors. In contrast, to ensure the clinical applicability of future FLASH RT across a wide range of tumors, including those that are deep seated and resistant to radiation, high dose rate beams of highly penetrating radiation will be required—none of the existing systems can deliver such beams [44]. Currently, clinical trials involving FLASH RT for deep-seated tumors may utilize scanned proton beams in transmission mode, i.e., without the Bragg peak. This approach involves covering the entire target volume, provided that the FLASH effect remains unaffected by the increase in LET in the Bragg peak or by the scanning of the beam [84]. Clearly, the clinical potential of proton beams is hindered by the cost and size of the installations, which require dedicated therapy centers. Given the constraints of photon beams in terms of maximum achievable dose rate, attention is now focused on very high-energy electrons (VHEEs), which have the potential to enable the treatment of deep-seated tumors while being compatible with more compact and affordable installations. Electron beams are unaffected by the loss of efficiency in bremmstrahlung electron/photon conversion. Consequently, power requirements for the accelerator could be up to three orders of magnitude lower. Additionally, given their sufficiently high energy, electrons can deposit energy at any given depth with minimal lateral diffusion. VHEEs, with energies ranging from 150 MeV to 250 MeV, exhibit attractive characteristics in terms of percentage depth dose (PDD) and lateral beam profiles [85]. This results in superior dose distribution compared to external megavoltage photon beams in specific Monte Carlo simulated treatment plans, such as those related to prostate, lung, or pediatric cancers. Indeed, a comparison [86] of treatment planning with photons, protons, and very high-energy electrons (VHEEs) has shown that VHEEs are capable of comparable if not superior performance in terms of dose conformation.

In light of the above, a combination of VHEEs and FLASH RT may offer a comprehensive solution for the clinical translation of this innovative approach. However, generating electron beams within the 150–250 MeV range with a compact footprint will demand advanced accelerator technology. Existing RF linac technology displays a low acceleration gradient, necessitating excessively large accelerator lengths. This results in clinical equipment of prohibitive size and cost, inevitably restricting access to future FLASH RT. Notably, this limitation has already impeded the progress of phase-contrast X-ray imaging based on synchrotron radiation, despite its promising performance in early cancer detection [87]. This technology is still awaiting translation from laboratory demonstration to clinical application. Hence, it is crucial to invest in the development of accelerator technologies that, in principle, can overcome these limitations and provide compact and affordable accelerators suitable for placement inside hospitals. High gradient accelerator systems based on the C-band technology [88] or on the X-band technology developed for a high-energy physics accelerator CLIC [89] are being considered for the design of a compact VHEE accelerator with UHDR capabilities. 

However, the energy required for VHEE RT, ranging between 100 and 250 MeV, might still be too high for a compact RF accelerator, even considering high gradient RF cavities. A disruptive approach based on laser plasma acceleration (LPA) is also being considered, which has no such limitations in terms of electron energy. LPA can easily provide VHEE beams [90] with a compact size and innovative setup based on optical technology rather than RF technology and can already deliver Gy doses per shot on a pencil beam-like configuration, with an instantaneous dose rate that can exceed by orders of magnitude the expected FLASH dose rates. Such pencil beams could be scanned to cover larger target volumes in a similar fashion as is currently performed with proton beams. Significant technological developments are still needed to reach the specifications of clinical FLASH-RT in terms of dose per fraction over a larger area for clinical treatment. The main approach here consists of increasing the average power and repetition rate of the driving laser power source from the current 10 W–10 Hz to 100 W–100 Hz that is currently being developed at an industrial level for this class of accelerators. Our roadmap at CNR-INO for the development of a clinical VHEE device is indeed based on a laser plasma accelerator setup, exploiting proof of principle experimental demonstrations of VHEE beam generation and dosimetry [91] and building on fast-developing laser technology [92]. Our program includes several milestones, the primary being VHEE beam control and stability, the repetition rate of beam operation for FLASH dose compliance, and the final assessment of source clinical readiness. Similar projects are ongoing at other main labs, like the LAPLACE center in Palaiseau or the EU-funded Ebeam4Therapy project at the Weizmann Institute of Science (WIS), while companies are also emerging that are aiming at the industrialization of such highly innovative concepts and the development of robust power laser systems. 

Longer-term developments include innovative laser technology with ceramic materials, which may enable even higher efficiency and a more compact accelerator footprint. It is clear, however, that both RF and laser-driven acceleration will require significant investments to develop accelerators capable of delivering VHEE beams with a high dose rate for clinical use. On the other hand, preclinical research is necessary to confirm the effectiveness of the FLASH effect and understand the underlying fundamental mechanisms, and the development of new treatment planning systems for clinical translation can already be carried out with existing experimental accelerators.

## 6. Conclusions

Nowadays, FLASH RT is considered one of the most promising revolutions in radiation oncology, placing itself at the intersection of technology, physics, and biology. The unique healthy tissue-sparing effect and, at the same time, the equivalent tumor response have already been identified in vivo for multiple organ systems, such as the lung, brain, skin, intestine, and blood, and even in the first human patient. However, the fundamental mechanisms behind the FLASH effect are yet to be fully elucidated. Therefore, additional experimental studies and clinical investigations are necessary to confirm the conditions for normal tissue sparing of FLASH RT. 

Better optimization of parameters and technological challenges is fundamental for making the clinical translation of FLASH RT feasible. From a biological point of view, there is an urgent need to further understand the impact on DNA and how this varies with UHDR irradiation and in the presence of other biological factors (e.g., hypoxia, tumor microenvironment). 

In silico methods can be valuable tools for designing in vitro studies on DNA damage, allowing for model distributions of the direct damage, as well as the diffusion and reaction of free radicals involved in the indirect action. 

Three-dimensional (3D) cell culture models and organoids can also provide significant insights into the investigation of the complex cellular response in tumors and normal tissues after UHDR radiation, representing a simplified model of the structures and functions of in vivo organs. 

This knowledge is essential for a better understanding of the FLASH-sparing effect of normal tissues and a more rapid translation to the clinic. 

In addition to these fundamental aspects, clinical translation is still hindered by the lack of accessible devices that can provide UHDR beams with therapeutic capabilities. Notable exceptions are low-energy, IORT-like electron beam accelerators that are being considered for the FLASH radiotherapy of skin cancer and hadron therapy centers that could provide UHDR beams in transmission mode for the future FLASH treatment of deep-seated tumors. A general approach to FLASH radiotherapy still relies on compact and affordable medical accelerators capable of UHDR irradiation for delivering FLASH radiotherapy. To this aim, it is clear that VHEE beams are an excellent potential solution based either on radiofrequency conventional accelerators or on the most innovative laser-driven plasma accelerators. In parallel, major developments are needed to develop treatment planning systems based on the actual or expected beam specifications of future VHEE clinical accelerators to ensure timely application of clinical protocols. Highly motivated developments are taking place in all these directions, providing a clear path to full clinical translation of FLASH radiotherapy.

## Figures and Tables

**Figure 1 ijms-25-02546-f001:**
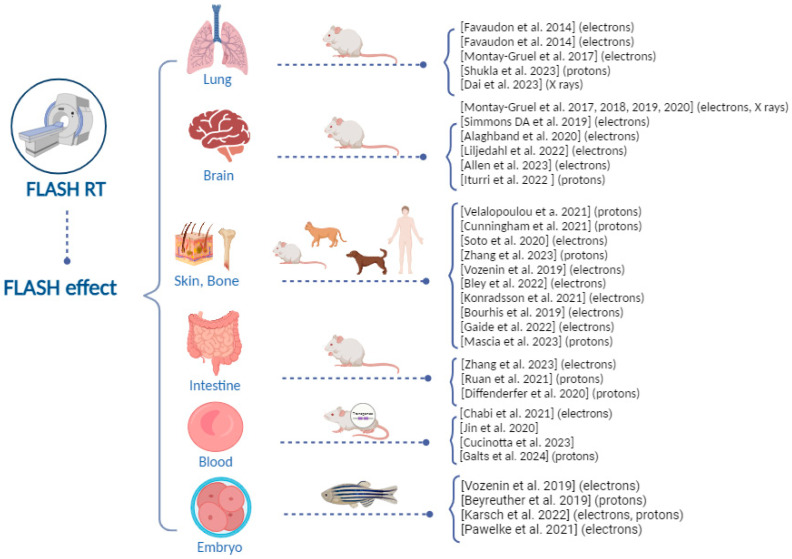
Overview of the preclinical and first-in-human clinical evidence about the ultra-high dose rate FLASH effect [1,14,15,16,17,18,19,20,21,22,23,24,25,26,27,28,29,30,31,32,33,34,38,39,40,41,42,43,44,45,47,48,50].

**Figure 2 ijms-25-02546-f002:**
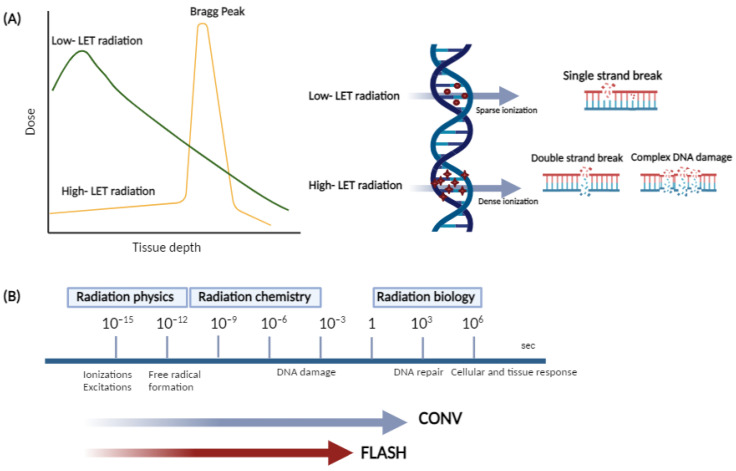
(**A**) Schematic illustration of depth dose distribution and DNA damage induction patterns for low and high LET beams; (**B**) timescales of physical, chemical, and biological phases of conventional and FLASH radiotherapy.

**Figure 3 ijms-25-02546-f003:**
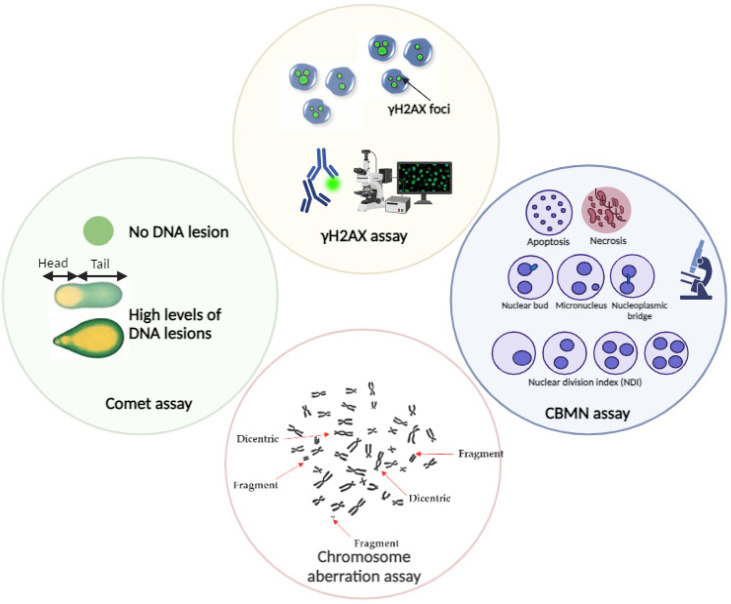
Schematic representation of the in vitro tests for ionizing radiation-induced DNA damage quantification.

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
