# Peer review of "FLASH Radiotherapy: Expectations, Challenges, and Current Knowledge"

_ijms, 2024, doi:10.3390/ijms25052546_

Round 1

Reviewer 1 Report

Comments and Suggestions for Authors

The work by Borghini et al. is a very comprehensive and a well-written review on the status of the FLASH therapy. The work is logically divided into chapters that describe the most important studies reporting tumour and normal tissue responses to FLASH radiation, treatment of human patients and first clinical trials, biological mechanisms beyond the FLASH effect and a chapter that focuses on the status of the technologies, which are/would be suitable for delivery of FLASH radiation.

Minor comments:

-          Part 2.3:  when referring to the publication by Vozenin et al. (ref. 32), it would be good to describe very shortly a little bit more in detail major findings of that study

-          Perhaps it would be instrumental to briefly elaborate on the (characteristics) of the various types of UHDR irradiations (proton, X-ray, electron) and perhaps also to summarize which studies were performed with which one (by adding this information to the figure for example)

-          Line 208 – the ‘0.08 Gy/s vs. 166 Gy/s’ should be assumingly ‘166 Gy/s vs. 0.08 Gy/s’?

-          Please add some references into subchapter 4.1 (albeit this subchapter refers to facts that are indeed very well established, there still should be indicated some literature (review, book) supporting the claims).

-          Please, add ‘seconds’ to the time-scale in Fig.2B.

-          Please, add (A) to the caption of Figure 2.

Comments on the Quality of English Language

-          there are some minor language mistakes and abbreviations are sometimes not spelled out at their first mention (for example CONV RT) or not always consequently used when already introduced

Author Response

The work by Borghini et al. is a very comprehensive and a well-written review on the status of the FLASH therapy. The work is logically divided into chapters that describe the most important studies reporting tumour and normal tissue responses to FLASH radiation, treatment of human patients and first clinical trials, biological mechanisms beyond the FLASH effect and a chapter that focuses on the status of the technologies, which are/would be suitable for delivery of FLASH radiation.

Thank you very much for your kind comments.

Minor comments:

-Part 2.3: when referring to the publication by Vozenin et al. (ref. 32), it would be good to describe very shortly a little bit more in detail major findings of that study

As suggested by the reviewer we have shortly described the major findings of the study by Vozenin et al. as follow:

“By using the radiation-induced depilation and skin fibrosis as acute and late endpoint respectively, a protective effect of FLASH-RT was observed in minipigs and cats [32]. Pig skin was irradiated to single-fraction radiation doses of 28, 31, or 34 Gy using either conventional  (0.083 Gy/second)or ultrahigh dose-rate electron therapy (300 Gy/s).  The presence of late effects such as fibronecrosis, collagen deposition, and skin contracture was greater in animals irradiated with conventional dose rates [32]. These preclinical results are consistent with results obtained in a veterinarian clinical trial conducted in cat patients with squamous cell carcinoma of the nose. Cats were treated using single fractions from 25 to 41 Gy at ultrahigh dose-rates. No dose-limiting toxicity and relatively mild long-term toxicity were found. FLASH-RT treatment yielded a favorable outcome with complete response at 3 months for all cat patients and a free-survival rate of 84% at 16 months [32].”

-Perhaps it would be instrumental to briefly elaborate on the (characteristics) of the various types of UHDR irradiations (proton, X-ray, electron) and perhaps also to summarize which studies were performed with which one (by adding this information to the figure for example)

Accordingly, we added this useful information to the Figure 1.

-Line 208 – the ‘0.08 Gy/s vs. 166 Gy/s’ should be assumingly ‘166 Gy/s vs. 0.08 Gy/s’?

-Please add some references into subchapter 4.1 (albeit this subchapter refers to facts that are indeed very well established, there still should be indicated some literature (review, book) supporting the claims).

-Please, add ‘seconds’ to the time-scale in Fig.2B.

-Please, add (A) to the caption of Figure 2.

We have modified the manuscript following the suggestions of the Reviewer.

Comments on the Quality of English Language

-there are some minor language mistakes and abbreviations are sometimes not spelled out at their first mention (for example CONV RT) or not always consequently used when already introduced

We have checked mistakes and abbreviations, and we have corrected them throughout the manuscript.

Reviewer 2 Report

Comments and Suggestions for Authors

- Author have reviewed moslty on the biological reports on FLASH effect and highlighted important issues. However, the review is basically summarized on ultra high dose rate, and without consideration on radiation types and other important parameters, such as radiation types, dose range, pulse width and intervals.  I feel its a bit out of date, since there are enourmous amout of reviews concerning FLASH effects, on both biological, physi-chemical aspect, and simulations.

I suggest that authors should consider including the above parameters in the reviews, or at least review in topics of beam parameters, and not just summarize in biological aspect.

- Authors have mentioned about the difference of time scale of UHDR and CONV exposure in Figure 2. But not about the time scales on radical recombination and oxygen depletion, which is very important to show how it is correlate to the time scales mentioned in Figure 2.

- The paper is sectionized in multiple topics, and I agree that the most of the  sub topics under 4. BIological mechanisms behind the FLASH effect, are important issues. It is well summarized, but I feel that review paper should have also, perspectives, problem/issues that should to be highlighted, with the authors conclusions. Theses points are very weak and should strengthen with the purpose/reason of this "review" paper.

- There are several mistypos, 106 should 10 6(upper letter), 10 8(upper letter) and so on.

Author Response

Author have reviewed moslty on the biological reports on FLASH effect and highlighted important issues. However, the review is basically summarized on ultra high dose rate, and without consideration on radiation types and other important parameters, such as radiation types, dose range, pulse width and intervals.  I feel its a bit out of date, since there are enourmous amout of reviews concerning FLASH effects, on both biological, physi-chemical aspect, and simulations.

I suggest that authors should consider including the above parameters in the reviews, or at least review in topics of beam parameters, and not just summarize in biological aspect.

Thank you for calling attention to this issue. We briefly discussed this point in line with the reviewer’s suggestion as follows:

In fact, additional information can be obtained by properly varying beam properties that, in principle, can act on the physical processes behind the FLASH effect. Here we expect that parameters like radiation type, temporal and spatial structure, dose range and energy spectrum are expected to play a role. Systematic studies of these dependencies are still lacking, but a number of studies are emerging which focus on specific radiation parameters. Among these parameters, the beam temporal structure was varied to investigate oxygen depletion in water [74], measuring O2 content for different average and bunch dose rates of electron beams, showing strong correlation with biological data, supporting the role of radicals at the origin of the FLASH effect. In another study [75], the dependence of O2 consumption and H2O2 production were found to depend on the mean dose rate, with instantaneous dose rates also contributing to this effect. Concerning the type of radiation, interesting studies are finally emerging on the use of UHDR kilovoltage (kV) X-rays from a rotating-anode x-ray source for in vivo studies, making this type of radiation easily accessible at a laboratory scale, compared to synchrotron radiation. A recent study [76] reported observation of FLASH effect in normal tissue sparing of radiation toxicities in mouse skin irradiated at 35 Gy with similar tumor growth suppression compared to conventional irradiation. Again concerning the type of radiation, a comprehensive comparison [77] of electron and proton irradiation with UHDR and CONV modalities showed that the neurocognitive capacity of both electron and proton FLASH irradiated mice was indistinguishable from the control, while both electron and proton CONV irradiated cohorts showed cognitive decrements. More specifically, normal brain protection was achieved when a single dose of 10 Gy was delivered in 90 ms or less, suggesting that the most important physical parameter driving the FLASH sparing effect might be the mean dose rate.”

- Authors have mentioned about the difference of time scale of UHDR and CONV exposure in Figure 2. But not about the time scales on radical recombination and oxygen depletion, which is very important to show how it is correlate to the time scales mentioned in Figure 2.

We thank the Reviewer for having raised this important point. Now, we added a brief description of the time scale (physical, chemical, and biological stages) of radiation damage.

“Typically, the radiolytic events occur in three main stages taking place on different typical time scales. During the first or “physical” stage, which takes place within  10−15−10−12s, extremely reactive free radicals (e.g. aqueous or hydrated electrons and other reactive oxygen species, such as H2O2, O2 or OH)   are produced and undergo fast reorganization in the chemical stage (10−12 −10−6 s), leading to the formation  of an array of  reactive products, which in turn can break the chemical bonds and produce DNA damage and possible repair processes in the cell over a wide time scale of  (“biological” stage). FLASH irradiation is around 1000 times faster than conventional irradiation and this might interfere with the radiation-chemical reactions, and consequently with the biological processes in response to irradiation.”

- The paper is sectionized in multiple topics, and I agree that the most of the  sub topics under 4. BIological mechanisms behind the FLASH effect, are important issues. It is well summarized, but I feel that review paper should have also, perspectives, problem/issues that should to be highlighted, with the authors conclusions. These points are very weak and should strengthen with the purpose/reason of this "review" paper.

This comment is really relevant and we have done our best to better discuss the physical challenges and perspectives in our conclusion as follows:

Besides these fundamental aspects, clinical translation is still hindered by the lack of accessible devices that can provide UHDR beams with therapeutic capabilities. Notable exceptions are the low energy, IORT-like electron beam accelerators that are being considered for FLASH radiotherapy of skin cancer, and hadron therapy centers that could provide UHDR beams in transmission mode for future FLASH treatment of deep seated tumors. A general approach to FLASH radiotherapy still relies on compact, affordable medical accelerators capable of UHDR irradiation for delivering FLASH radiotherapy. To this aim it is clear that VHEE beams are an excellent potential solution, based either on radiofrequency conventional accelerators or on the most innovative laser-driven plasma accelerators. In parallel, major developments are needed to develop treatment planning systems based on the actual or expected beam specifications of future VHEE clinical accelerators, to ensure timely application of clinical protocols. Highly motivated developments are taking place in all these directions, providing a clear path to full clinical translation of FLASH radiotherapy.

- There are several mistypos, 106 should 10 6(upper letter), 10 8(upper letter) and so on.

 The grammar errors were corrected appropriately.

Reviewer 3 Report

Comments and Suggestions for Authors

The review "FLASH radiotherapy: expectations, challenges, and current knowledge" submitted by A. Borghini and colleagues aims to provide an "...overview of the research progress of FLASH-RT and discuss the potential challenges to be faced before its clinical applications become a reality" (Abstract, lines 18-20). This goal is not readily achieved and the paper requires serious revision in many parts.

None of the comprehensive reviews on FLASH-RT published recently in the FLASH special issue of Medical Physics (Med Phys 49 [2022] 1972-2095), the International Journal of Radiation Biology (Kacem et al. Int J Radiat Biol 98 [2022] 506-16) and the Annual Review of Cancer Biology (Limoli & Vozenin Annu Rev Cancer Biol 7 [2023] 1-21) is quoted here. The authors are urged to revise their MS in light of these papers.

Abstract, lines 11-13:

(a) "Most animal studies suggest" should read "All animal studies have shown"

(b) "sparing effect without modifying damage to cancer cells" should read "sparing effect in normal tissues without modifying tumor cure in animal models"

(c) Remark on "only recently preclinical and first-in-human trials are emerging": the first preclinical evidence and the first in-human trial were published in 2014 and 2019, respectively.

Abstract, lines 15-16:

Remark on "Suitable in vitro studies are required to fully understand the radiobiological mechanisms associated with UHDR". Right now, all in vitro studies (cells in culture) have failed to provide any decisive information on the FLASH mechanisms, due likely to bad methods and poor control of metabolic conditions.

Page 1, line 34: delete [1,2] and quote here the founding paper [15].

Page 1, line 42: delete [4,5] and quote the original (experimental) paper [15] and comprehensive reviews, e. g. Friedl et al. Med Phys 49 [2022] 1993-2013 and Limoli & Vozenin Annu Rev Cancer Biol 7 [2023] 1-21.

The contribution of the immune system to the FLASH effect (p. 2, lines 76-84) has been the object of a contentious debate in the course of the FRPT 2023 meeting (Toronto, December 5-7). See also page 10 in Limoli & Vozenin 2023.

Page 4, lines 157-167. Contradictory results have actually been reported. The problem probably lies in dosimetry or, more exactly, in differences in the size of the irradiated volumes or depth-dose distribution among different beams. In addition to references 11, 34 and 35, please quote and discuss Diffenderfer et al. IJROBP 106 [2020] 440-8; Kim et al. Cancers (Basel) 13 [2021]; Zhu et al. Med Phys 49 [2022] 4812-22; Shi et al. PNAS 119 (2022) e2208506119; Eggold et al. Mol Cancer Ther 21 (2022) 371-81.

Paragraph 4 "Biological mechanisms beyond the FLASH effect" encompasses 4 pages from line 235 to line 414 and focuses onto DNA damage. This section should be reduced to one single page and revised in-depth with the aid of scientists well-trained in DNA repair pathways.

- Old habits die hard, and so does the Transient Oxygen Depletion (TOD) model. Direct, time-resolved measurements of oxygen in cells or tissues have conclusively shown that oxygen depletion during pulse irradiation at ultrahigh dose rate is marginal and cannot account for the FLASH effect in healthy, normoxic tissue (Cao et al. IJROBP 111 [2021] 240-8; Jansen et al. Med Phys 48 [2021] 3982-90; El Khatib et al. IJROBP 113 [2023] 624-34).

- No difference in the incidence of gamma-H2AX foci was observed (i) between normal vs. tumor cells and (ii) between FLASH vs. conventional dose rate (Fouillade et al. Clin Cancer Res 26 [2020] 1497-506). Of note, at 37°C gamma-H2AX foci peak at 30 min post-irradiation; at that time, 50% of DNA double-strand breaks have been rejoined by NHEJ.

- Recently, Barghouth et al. (Radiother Oncol 188 [2023] 109906) demonstrated that FLASH-RT and CONV-RT are indistinguishable in the production of chromosomal junction structures, independently on the concentration of oxygen.

The authors are more familiar with the subject matter of paragraph 5. It should be mentioned that the feasibility of X-rays at FLASH dose rates has been demonstrated (Sampayan et al. Sci Rep 11 [2022] 17104 and others), albeit at a prohibitive cost. The possibility and promise of VHEE is well presented; Fallace et al. Phys Med 104 [2022] 149-51 should be quoted. In contrast, the future of laser plasma machines is open to question for technical reasons (relatively low dose per pulse, low repeat frequency, small size of the beams making it necessary to use spot-scanning).

 Conclusion, line 529: cell cultures and organoids are not good models for FLASH studies. However, ex vivo approaches like tissue slices, have been successfully tested.

Comments on the Quality of English Language

Minor editing required.

Author Response

The review "FLASH radiotherapy: expectations, challenges, and current knowledge" submitted by A. Borghini and colleagues aims to provide an "...overview of the research progress of FLASH-RT and discuss the potential challenges to be faced before its clinical applications become a reality" (Abstract, lines 18-20). This goal is not readily achieved and the paper requires serious revision in many parts.

None of the comprehensive reviews on FLASH-RT published recently in the FLASH special issue of Medical Physics (Med Phys 49 [2022] 1972-2095), the International Journal of Radiation Biology (Kacem et al. Int J Radiat Biol 98 [2022] 506-16) and the Annual Review of Cancer Biology (Limoli & Vozenin Annu Rev Cancer Biol 7 [2023] 1-21) is quoted here. The authors are urged to revise their MS in light of these papers.

Thank you for reviewing our manuscript and for your valuable comments. Currently, FLASH-RT has generated significant interest in both clinical and research communities, and we are aware that there is an abundance of excellent reviews in the field of FLASH-RT.  Overall, this paper provides an overview of the current research on the FLASH effect and the clinical promise of this modality for cancer treatment, with a particular focus on the importance of studying the DNA effects following  UHDR irradiation, which are yet largely unexplored.  This is particularly interesting in light of the fact that DNA is the main target of radiation therapy. Therefore, more research into the basic biology of DNA damage induction and levels, as well as on the subsequent DNA repair cellular efficiency is needed to provide insights into the FLASH effect. We believe that this issue is of great interest to readers and an important research area. Anyway, we understand your concern about the inclusion of these comprehensive recent reviews on FLASH-RT. We have done our best to improve the readability and to add the reviews published recently in the FLASH special issue of Medical Physics. The references added in the paper are listed below:

Farr, J.B.; Parodi, K.; Carlson, D.J. FLASH: Current status and the transition to clinical use. Med. Phys. 2022, 49, 1972-1973.

Limoli, C.L.; Vozenin, M.-C. Reinventing radiobiology in the light of FLASH radiotherapy. Annu. Rev. Cancer Biol. 2023, 7, 1–21.

Kacem H, Almeida A, Cherbuin N, Vozenin MC. Understanding the FLASH effect to unravel the potential of ultra-high dose rate irradiation. Int J Radiat Biol. 2022, 98, 506-516.

Abstract, lines 11-13:

(a) "Most animal studies suggest" should read "All animal studies have shown"

(b) "sparing effect without modifying damage to cancer cells" should read "sparing effect in normal tissues without modifying tumor cure in animal models"

(c) Remark on "only recently preclinical and first-in-human trials are emerging": the first preclinical evidence and the first in-human trial were published in 2014 and 2019, respectively.

We have revised the paper according to your suggestions. See changes in red in the abstract.

Abstract, lines 15-16:

Remark on "Suitable in vitro studies are required to fully understand the radiobiological mechanisms associated with UHDR". Right now, all in vitro studies (cells in culture) have failed to provide any decisive information on the FLASH mechanisms, due likely to bad methods and poor control of metabolic conditions.

You are right. In general, more preclinical studies are needed. We believe that in vitro studies are crucial to define the optimal beam parameters and the radiobiological mechanisms involved in the FLASH effect.  Specifically, we believe that cytogenetic tests (e.g dicentric chromosome analysis, cytokinesis-block micronucleus assay which are considered the gold-standard in radiobiology to  estimate biological effects of absorbed dose) are crucial  for investigating the cell response to  UHDR and how the FLASH effect in human cells depends on the physical aspects of the radiation beams and the pulse-related factors, such as dose and dose rate per pulse, as well as the frequency of the pulse delivery. Therefore, in vitro studies play a crucial role in complementing in vivo studies in this field of research, allowing the manipulation of specific parameters with precision and control with respect to in vivo studies as well as studying specific cellular mechanisms and understanding the basic biology underlying complex physiological processes.

Page 1, line 34: delete [1,2] and quote here the founding paper [15].

Page 1, line 42: delete [4,5] and quote the original (experimental) paper [15] and comprehensive reviews, e. g. Friedl et al. Med Phys 49 [2022] 1993-2013 and Limoli & Vozenin Annu Rev Cancer Biol 7 [2023] 1-21.

The contribution of the immune system to the FLASH effect (p. 2, lines 76-84) has been the object of a contentious debate in the course of the FRPT 2023 meeting (Toronto, December 5-7). See also page 10 in Limoli & Vozenin 2023.

Page 4, lines 157-167. Contradictory results have actually been reported. The problem probably lies in dosimetry or, more exactly, in differences in the size of the irradiated volumes or depth-dose distribution among different beams. In addition to references 11, 34 and 35, please quote Diffenderfer et al. IJROBP 106 [2020] 440-8; Kim et al. Cancers (Basel) 13 [2021]; Zhu et al. Med Phys 49 [2022] 4812-22; Shi et al. PNAS 119 (2022) e2208506119; Eggold et al. Mol Cancer Ther 21 (2022) 371-81.

Thank you for these helpful comments. Accordingly, we quoted most of the suggested papers.

Paragraph 4 "Biological mechanisms beyond the FLASH effect" encompasses 4 pages from line 235 to line 414 and focuses onto DNA damage. This section should be reduced to one single page and revised in-depth with the aid of scientists well-trained in DNA repair pathways.

We are reluctant to follow this suggestion because one of the main focuses of this paper is the DNA damage following UHDR to improve the understanding of FLASH effect, according to our long-standing experience in the field of DNA-radiation effects in medical imaging.

- Old habits die hard, and so does the Transient Oxygen Depletion (TOD) model. Direct, time-resolved measurements of oxygen in cells or tissues have conclusively shown that oxygen depletion during pulse irradiation at ultrahigh dose rate is marginal and cannot account for the FLASH effect in healthy, normoxic tissue (Cao et al. IJROBP 111 [2021] 240-8; Jansen et al. Med Phys 48 [2021] 3982-90; El Khatib et al. IJROBP 113 [2023] 624-34).

- No difference in the incidence of gamma-H2AX foci was observed (i) between normal vs. tumor cells and (ii) between FLASH vs. conventional dose rate (Fouillade et al. Clin Cancer Res 26 [2020] 1497-506). Of note, at 37°C gamma-H2AX foci peak at 30 min post-irradiation; at that time, 50% of DNA double-strand breaks have been rejoined by NHEJ.

You are right. The work by Cao and co-workers suggested that oxygen depletion during pulse irradiation at an ultrahigh dose rate is marginal and cannot account for the FLASH effect in healthy, normoxic tissue. However, the oxygen’s role in the ‘FLASH effect’ requires further proof and investigation, as acknowledged by the same authors. As we discussed in the paper, a more recent study by Cooper and co-workers  (Br J Radiol 2022)  showed,  using the alkaline comet assay, that lower levels of DNA damage are induced following FLASH irradiation, an effect that is modulated by the oxygen tension, indicating that an oxygen related mechanism (e.g. transient radiation-induced oxygen depletion) may contribute to the tissue sparing effect of FLASH irradiation.

- Recently, Barghouth et al. (Radiother Oncol 188 [2023] 109906) demonstrated that FLASH-RT and CONV-RT are indistinguishable in the production of chromosomal junction structures, independently on the concentration of oxygen.

We would like to thank the Reviewer for having indicated this important recent finding. This information was added in the new version.

By using a high-throughput genome-wide translocation sequencing approach, a very recent study found no significant differences in the decrease in translocations or alteration of junction structures between FLASH-RT and CONV-RT in human embryonic transformed kidney cell line (HEK239T cells), across a wide range of oxygen tensions.  This suggests that both modalities of dose delivery induce similar DNA damage response in the in vitro model under investigation, independently on the concentration of oxygen.

The authors are more familiar with the subject matter of paragraph 5. It should be mentioned that the feasibility of X-rays at FLASH dose rates has been demonstrated (Sampayan et al. Sci Rep 11 [2022] 17104 and others), albeit at a prohibitive cost. The possibility and promise of VHEE is well presented; Fallace et al. Phys Med 104 [2022] 149-51 should be quoted. In contrast, the future of laser plasma machines is open to question for technical reasons (relatively low dose per pulse, low repeat frequency, small size of the beams making it necessary to use spot-scanning).

Conclusion, line 529: cell cultures and organoids are not good models for FLASH studies. However, ex vivo approaches like tissue slices, have been successfully tested.

The organoids, three-dimensional in vitro structures that resemble the tissue of origin,  represent an ideal in vitro research model to study the radiosensitivity and radiation damage of normal tissues (e.g. doi: 10.3390/ijms241310620; doi: 10.3389/fonc.2022.888416; 10.1007/s12195-020-0062). The application of organoids in FLASH RT is yet limited, but these models  can be employed as a powerful tool to explore  the interaction of different cell types under FLASH radiation as well as to study DNA damage, inflammatory and metabolic response after FLASH RT.

Round 2

Reviewer 2 Report

Comments and Suggestions for Authors

Authors have revised the manuscript extensively. I believe that the review paper is now informative with novelness. Also they have concluded the paper clearly.

Author Response

Thank you very much for your kind words about our revised manuscript.

Reviewer 3 Report

Comments and Suggestions for Authors

Section 4 " Biological mechanisms behind the FLASH effect" should be revised starting from published experimental evidence invalidating the oxygen depletion model in normoxic tissues and reduced to less than 1 page. The role of the Von Hippel Lindau system should be taken into consideration to explain the effect of hypoxia on radiation response. At the present time there is no convincing evidence in favor of dose-rate dependent differential chromosome damage in cells and tissues, and simple chemical systems such as plasmids in the test tube, are not representative of physiological/metabolic  conditions.

The authors should avoid speculative considerations and focus onto Sections 3 and 5.

Comments on the Quality of English Language

Correct. Minor editing required in a few places.

Author Response

Section 4 " Biological mechanisms behind the FLASH effect" should be revised starting from published experimental evidence invalidating the oxygen depletion model in normoxic tissues and reduced to less than 1 page. The role of the Von Hippel Lindau system should be taken into consideration to explain the effect of hypoxia on radiation response.

With regard to the experimental evidence invalidating oxygen depletion, we addressed this issue by citing recent works (Cao et al. IJROBP 111 [2021] 240-8; Jansen et al. Med Phys 48 [2021] 3982-90; El Khatib et al. IJROBP 113 [2023] 624-34). Their studies, indeed, suggest that oxygen depletion during pulse irradiation at an ultrahigh dose rate is marginal and cannot entirely account for the FLASH effect in healthy normoxic tissues. Accordingly, we modified our manuscript (Section 4) as follows:

“However, even though one of the most widely considered hypotheses being that the effect is related to substantial oxygen depletion upon FLASH, recent observations showed that oxygen depletion during pulse irradiation at ultrahigh dose rate is marginal and cannot entirely account for the FLASH effect in healthy normoxic tissues [49-51].”

At the present time there is no convincing evidence in favor of dose-rate dependent differential chromosome damage in cells and tissues, and simple chemical systems such as plasmids in the test tube, are not representative of physiological/metabolic  conditions. The authors should avoid speculative considerations and focus onto Sections 3 and 5.

Plasmid DNA serves as a convenient and simplified model for studying the DNA-damaging effects of ionizing radiation, enabling the assessment of primary DNA lesion yields. However, research involving plasmids falls short in providing a precise understanding of the cellular response mechanisms to ionizing radiation. In contrast, the assessment of chromosome aberrations involves at least two intertwined processes—lesion induction and subsequent lesion repair. Unexpectedly, there is limited data available on the impact of UHDR on chromosomal damage. This is particularly surprising considering that cytogenetic tests, including MN and dicentric assays, are considered the gold standard in radiobiology, as acknowledged by the International Atomic Energy Agency (IAEA) in their Technical Reports Series No. 405, titled "Cytogenetic Analysis for Radiation Dose Assessment" (IAEA, Vienna, 2001). Accordingly, one of the main objectives of this review is precisely to underline the importance of using standard tests to obtain information on the FLASH effect.